# Nutritional Nesting (Nestrition): Shaping the Home Food Environment in the First Pregnancy

**DOI:** 10.3390/nu16193335

**Published:** 2024-09-30

**Authors:** Chagit Peles, Netalie Shloim, Mary C. J. Rudolf

**Affiliations:** 1Azrieli Faculty of Medicine, Bar-Ilan University, Safed 1311502, Israel; mary.rudolf@biu.ac.il; 2School of Healthcare, University of Leeds, Leeds LS2 9JT, UK; n.shloim@leeds.ac.uk

**Keywords:** first pregnancy, nesting, prenatal nutrition, nutrition education, home food environment

## Abstract

Objective: To investigate primiparous women’s knowledge, attitudes, and practices regarding the physical home food environment (PHFE) and to assess if the first pregnancy provides a teachable opportunity to enhance the PHFE of first-time pregnant couples. Design: Longitudinal in-depth qualitative study involving questionnaires and individual interviews during and after pregnancy. Participants: Fifteen primigravida women. Main outcome measures: Knowledge, attitudes, and behaviors concerning PHFE; lifestyle and dietary habits; and interest in guidance regarding healthy PHFE during the first pregnancy and the transition to motherhood. Analysis: Thematic analysis for qualitative data and descriptive statistics. Results: Key findings include the significance of health, nutrition, and spousal support in the transition to motherhood. The first pregnancy was recognized as a critical period for establishing a healthy PHFE, while noting the physical and emotional challenges involved. A gap was found in guidance regarding PHFE for first-time pregnant women despite their interest in practical advice. Conclusions and Implications: The first pregnancy presents a significant opportunity to improve PHFE. ‘Nestrition’ (nutritional nesting), a new health-promotion strategy, incorporates nutrition education to enhance this process. These data support future research encompassing partners and diverse at-risk populations prior to the development of effective nutrition education for PHFE in the first pregnancy.

## 1. Background

A growing body of evidence suggests that in utero conditions during critical periods of developmental plasticity result in lifelong effects on offspring and may program obesity in later life [1]. These conditions highlight the relationship between nutrition factors in early life and fetal programming for adult diseases [2]. Importantly, most of these factors are potentially modifiable and therefore fundamental targets for obesity prevention efforts [3,4]. Particular emphasis needs to be placed on nulliparous mothers, as their offspring have higher risks of childhood overweight and adverse metabolic profiles [5]. Despite this, inadequate focus is given to modifiable prenatal factors influencing the development of child obesity [6,7].

‘Human nesting’ is one of the surrounding elements influencing pregnancy. It is defined as measurable changes in behavior and attitudes related to birth and safe environment preparation that occur during pregnancy [8]. Nesting includes aspects of space preparation, familiarity preference, and novelty aversion starting at the end of pregnancy and extending into the first months postpartum [9]. The urge to nest is a powerful motivating force despite the common experience of significant fatigue in the third trimester. Yet there is limited literature directed at human nesting behaviors [10].

This study explores perceptions of a healthy home food environment (HFE) and is grounded in well-established theoretical foundations, namely the home food environment pertaining to childhood obesity [11], influences of microscale-built environments on food intake [12], and the environmental factors related to dietary disease risk [13]. The food environment pertaining to childhood obesity is composed of both physical and social domains, each with macro (community) and micro (home) contributions, together with the person-centered HFE [14]. The physical home food environment (PHFE) is the most proximal environment influencing food intake of special relevance to children. Examining ‘food landscapes’, focusing on availability (home food inventory) and accessibility (food access and placement in the home), and assessing individuals’ perceptions of their physical environment can shape the physical foundation of the HFE where new families are generated.

Pregnancy, and particularly the first pregnancy, may be a potential ‘teachable opportunity’ [15], a time when women are engaged in health care services and receptive to health messages, posing the question of whether the first pregnancy might be an opportune time for nutrition education. Pregnant women perceive pregnancy-specific nutrition information to be important because it is one of the few aspects that they can apply in their daily lives to protect the health of their baby [16]. Still, the number and character of health behaviors women may need to initiate or modify during pregnancy need to be considered when developing services targeting pregnant women [17,18]. There is a need for research addressing health behavior change during pregnancy in general and in the first pregnancy in particular.

Research on first-time pregnancy centers on labor and maternal health, with special attention paid to the needs of women with obesity or women living in poverty or who are otherwise marginalized [19]. There has been little research on tackling childhood obesity and improving health-related lifestyles through the PHFE in the first pregnancy, which is of relevance for practical interventions for new parents-to-be. 

This paper is the first in a series of articles involving both Israeli and English subjects and the baby’s father. The aim of this study was twofold. First, the aim was to examine the knowledge, attitudes, and practices of primiparous women pertaining to the PHFE. Second, the aim was to explore if a first pregnancy provides a teachable opportunity to enhance the PHFE of first-time pregnant couples. 

Gaining a deeper understanding of beneficial ways of establishing a healthier PHFE could significantly contribute to preventing obesity in children and parents [20,21]. The present study investigates whether preparing for a healthy PHFE among first-time pregnant families could offer a substantial opportunity for effective ‘up stream’ primary prevention.

## 2. Methods

The research is a longitudinal qualitative study focusing on first-time pregnant women conducted between 2020 and 2022. Primigravida women were interviewed individually following completion of an online questionnaire to ascertain their PHFE attitudes and practices. The interviews were conducted face to face by the first author (PC) in the participant’s home and took approximately an hour. The same fifteen participants, as new first-time mothers, were then interviewed face to face for a second time and completed a follow-up online questionnaire to determine change since birth. The purpose of this second stage was to explore participants’ newfound perspectives and practices related to their PHFE. In this study, the term ‘pregnant women’ encompasses all individuals capable of pregnancy, including transgender men and non-binary persons [22].

## 3. Recruitments and Participants 

Pregnant women were recruited using advertisements posted on social media websites and ‘WhatsApp’ groups designed for pregnant women, their partners, and their dietitians, and other pregnancy sites. Printed flyers were distributed at the host university, local libraries, and conferences held at the local hospital. Inclusion criteria consisted of being a first-time expectant over the age of 18. 

### 3.1. Study Materials


**Semi-structured interviews. *n* = 15 in pregnancy; *n* = 14 post-pregnancy.**


Individual face-to-face interviews were conducted in the second and third trimesters and again 5–18 months after birth using a semi-structured interview guide developed by the authors, who all have extensive clinical and research experience related to nutrition during pregnancy (see Appendix A). The interviews aimed to encourage participants to raise relevant issues within a structured framework in order to reach an understanding of their attitudes, perceptions, and motivations [23]. The topics included were the following: first pregnancy experience and support, perceived and actual PHFE, barriers to and levers for maintaining a healthy PHFE, whether this period of life may be a ‘teachable moment’ for preparing a healthy PHFE, and views on PHFE classes. Following a brief, informal enquiry into the wellbeing of the mother (and, in the second interview, the baby), interviews were conducted in a similar way with the same interview guide at both time points. Data were collected until saturation was reached [24]. This was defined as the point when no new themes emerged regarding primiparous women’s knowledge, attitudes, and practices and the potential of a first pregnancy as a teachable opportunity. Interviews were conducted and recorded and the transcription was translated by the lead researcher (PC), who is a clinical dietitian with 25 years of experience with women and mothers of preschool children and is a mother of a large family herself (see Appendix A). Transcription was undertaken by an individual unconnected to the research.


**Questionnaires. *n* = 15 in pregnancy; *n* = 15 post-pregnancy.**


A questionnaire (see Appendix A) was sent to the participants prior to their interviews in both phases of the study for completion online. The questionnaire was derived from validated questionnaires that address differing PHFE components [25,26] and did not differ at the two time points. It comprised 31 items and took about 20 min to complete. Questions were related to food preparation practices; food inventory relevant to childhood obesity, including fruits and vegetables, whole grains, high-fat foods, snacks, and beverages; kitchen furniture and appliances; platescapes; gardens; media equipment in dining areas; partners’ perceptions regarding PHFE; food shopping; breastfeeding intention; and nutrition program interest and format. The Food Frequency Questionnaires (short FFQ) [27] included foods related to risk and protective factors for childhood obesity in pregnant women’s diets. Participants provided sociodemographic information including religion or belief (no religion, Christian, Hindu, Jewish, Muslim, Sikh, other), postcode, marital status (married, living with partner, long-term relationship, single, divorced, other), and monthly household income before tax (ranging from less than GBP 500 to more than GBP 5000, with a ‘Prefer not to say’ option). Medical conditions were reported, with options including no conditions, diabetes, obesity, high blood pressure, or other specified conditions. Educational level was categorized from ‘No formal schooling’ to ‘Postgraduate degree’. Household food insecurity was measured by asking ‘The food we bought didn’t last, and we didn’t have money for more’ and ‘We worried food would run out before we got money to buy more’ (response options: ‘Often true, Sometimes true, Never true, Don’t know’) [28]. Body Mass Index (BMI) was calculated using self-reported height and weight at two time points. Additional details are provided in Table 1.

**Data analysis.** Our analysis followed the steps outlined by Braun and Clarke, 2006 [29] for conducting thematic analysis, which is a theoretical and foundational method of analyzing qualitative data sets that allows for flexible identification and categorization of patterns of experience or meaning. Our analysis was inductive and conducted at the latent level, seeking to identify the underlying meanings, assumptions, and patterns in women’s accounts. Terminologically, the term ‘expectants’ refers to participants during pregnancy, while ‘mothers’ denotes participants in the postpartum phase. The second set of interviews, conducted postpartum, was designed to evaluate how participants’ initial perceptions and practices concerning the PHFE shifted following childbirth. This comparative framework enabled a nuanced examination of both continuity and change in attitudes and practices as participants transitioned from pregnancy to motherhood, offering a comprehensive understanding of the unique challenges and opportunities that arise at each stage.

Our methodological approach was thoroughly documented and included the following stages. 1. Becoming familiar with the data: The researcher who is fluent in both languages translated all interviews into English, and the resulting transcripts were analyzed and annotated by the first author to identify the prominent themes or meanings evident in the data. To ensure reliability and depth of analysis, the second author, an expert in qualitative research, reviewed a randomly selected subsample of six interviews at each time point (12 in total). This subsample size was chosen to provide a manageable yet representative cross-section of the data, thus allowing for thorough verification without being overly burdensome. According to Guest, Bunce, and Johnson [30], a sample size of six interviews is often sufficient to achieve thematic saturation in qualitative research, thus ensuring that the most significant patterns and themes are captured. The subsample was reviewed and discussed until full agreement and consistency were reached between the authors. 2. Coding: In this stage, descriptive labels or codes were assigned to the data to indicate their potential or transparent meaning, which enabled the preliminary organization of the data. The coding process took place transcript by transcript and line by line, with the goal of identifying and grouping units of text that address the same issue or meaning under a common code. The codes were refined, merged, and expanded iteratively until all relevant aspects of the data pertaining to the research question were incorporated. The first author completed this stage for the entire analysis. The second author repeated the same process with a subsample of interviews. 3. Generating Themes: Following the combination of codes from the entire data set, the analysis proceeded to identify commonalities or shared relevance between the codes. These similarities were then clustered together to create preliminary themes that were more interpretive and conceptual in nature. The first author produced these, and they were subsequently refined through discussions with the other authors. At this stage, all data were reviewed, discussed, and agreed on by all authors. 4. Reviewing the Themes: During this stage, the themes’ validity was evaluated by examining their coherence with the data and the degree to which certain aspects of the data (whether newly discovered or not) lead to the refining of themes. 5. Defining and Naming Themes: The definitive set of themes was named, and a description of the core characteristics was written. Finally, all of the authors discussed the analytic output and its grounding in the data. 

**Quantitative analysis.** The median and interquartile range (IQR) were calculated for participants’ characteristics, as well as the mean and standard deviation for the FFQ. The FFQ analysis focused on fruit, vegetables, whole grains, and legumes as indicators of a healthy diet and high-fat, processed food, highly sugary food, processed meat, crisps, and sweet beverages as indicators of a less healthy diet (in line with modifiable risk factors for childhood obesity [13,31]). In view of the small sample size, a statistical comparison of measures at the two time periods was not conducted.

**Table 1 nutrients-16-03335-t001:** Participants’ characteristics (*n* = 15).

Participant			During Pregnancy					Post-Pregnancy				
	Pre-Pregnancy BMI (kg/m^2^)	Education ^1^ (1–6)	Age (Years)	Income ^2^ ($/Month)	GA (Weeks)	Healthy Lifestyle ^3^ (1–10)	Health Satisfaction ^4^ (1–5)	Income ^2^ (GBP/Month)	Mother’s BMI (kg/m^2^)	Baby’s Age (Month)	Healthy Lifestyle ^3^ (1–10)	Health Satisfaction ^4^ (1–5)
W.A	25.5	6	23	-	15	6	4	2.7 k	28.5	5	3	2
W.B	23.5	5	23	-	16	9	4	5.4 k	24	5	8	3
W.C	23	6	27	2.7 k	22	4	2	5.4 k	24	7	5	3
W.D	19	5	23	0.7 k	24	8	4	1.3 k	18.5	6	6	3
W.E	28.5	6	23	2.7 k	24	6	2	2.7 k	29	11	6	3
W.F	27.5	5	26	2.7 k	26	6	3	1.3 k	29	7	7	2
W.G	18.4	5	23	1.3 k	26	7	4	1.3 k	18.5	7	8	4
W.H	22	5	24	2.7 k	29	7	3	1.3 k	22	8	6	3
W.I	29.8	5	30	2.7 k	31	5	3	5.4 k	30.5	18	7	3
W.J	22	6	23	2.7 k	33	6	4	2.5 k	22	7	7	5
W.K	22	5	28	5.4 k	33	6	3	5.4 k	22.5	12	6	3
W.L	19	5	23	2.7 k	33	5	3	2.7 k	19	11	7	4
W.M	21	5	28	2.3 k	37	6	4	5.4 k	21	18	4	4
W.N	23	5	28	2.7	37	8	4	2.7 k	23	12	8	4
W.O	20	5	27	5.4 k	37	4	4	5.4 k	20	13	4	4
Mean (SD)	22.9 (3.5)	NA	25.3 (2.5)		28 (7.2)	6.2 (1.4)	3.4 (0.7)		23.4 (4.0)	9.8 (4.2)	6.1 (1.5)	3.3 (0.8)
Median	22	NA	24		29	6	4		22.5	8	6	3
IQR	20.5–24.5	NA	23–27.5		23.5–33	5.5–7	3–4		20.5–26.3	7–12	5.5–7	3–4

^1^ ‘What is the highest level of education you completed?’ … 5 = High school. 6 = College/University. ^2^ ‘What is your monthly household income before tax and including benefits (converted to $)?’ ^3^ ‘Overall, how healthy do you think your lifestyle is at the moment?’ 1 = not at all to 10 = very healthy [32]. ^4^ ‘How satisfied are you with your health?’ 1. Very dissatisfied to 5. Very satisfied [33].

### 3.2. Ethical Approval

This study received full ethical approval from the Faculty of Medicine Ethics Committee (Approval No. IRB09-2019), and written informed consent was obtained from all participants.

## 4. Results

### 4.1. Participants’ Characteristics (See Table 1)

All of the participants were married to male partners. Their mean age was 25.3 ± 2.5 years, and the average gestation was 28 ± 7 weeks. All had completed high school and were healthy according to self-reports of medical conditions, and four reported BMI > 25. Follow-up took place 4.6–18.2 months postpartum. Over the course of the study, income levels tended to increase to average and health satisfaction declined. 

There were observable changes in dietary consumption, though the small sample size precluded formal statistical analysis. Trends indicate that during pregnancy, most participants consumed more than five portions of fruits and vegetables daily (see Table 2). Postpartum, there was a decline in fruit and vegetable consumption alongside an increase in both coffee and low-calorie drink intake. Concurrently, there was a reduction in the presence of juices, candies, high-fat and sugary foods, processed meats, snacks, and sweet beverages within the home food environment.

### 4.2. Thematic Analysis

Four themes relating to the physical HFE emerged from the coding of the interviews: opportunities for improving the HFE; HFE-related challenges and difficulties; insufficient HFE-related guidance; and the narrow window of opportunity for change (see Table 3).

**THEME 1: First Pregnancy as an Opportunity to Improve the Home Food Environment.** Most participants during pregnancy and once they were mothers considered the first pregnancy as providing a narrow window of opportunity for appropriate guidance regarding the food environment at home. Despite the physical and emotional difficulties accompanying the first pregnancy, they remarked on relatively high motivation and availability for the establishment of a healthier home food environment. It is important to note that participants stressed that this motivation is relative to their past and present circumstances (see Table 3).

Iterating their greater time-bound motivation, participants stressed the significance of implementing a healthy HFE in advance of childbirth and doing so to ensure that their children will be born into a healthier environment, thereby granting an optimal starting point in life. They remarked that in the first pregnancy there is more time and greater motivation to form healthier habits that will persist. They added that it would make it easier for pregnant women and mothers to feel better and come to breastfeeding and baby care with a healthy lifestyle.

‘It’s best during the pregnancy—when a big change is going to happen. It’s a time when we start thinking not only about ourselves but also about the child that is going to be born and we want to be sure that he will have a life as healthy as possible so far as it concerns us as parents. In pregnancy, there is much more desire and willingness to change…’(W.D; 24 w pregnant)

Several reasons were offered by the interviewees for seeing their first pregnancy as a good opportunity to incorporate a healthier food environment at home, namely the significance of a healthy pregnancy, the significance of nutrition, specifically, and cooperation/support from their partners during the pregnancy.

**Importance of Pregnancy.** The women indicated that during pregnancy they became more attentive to their body’s needs and tried to maintain their health as much as possible. According to most, the high health awareness stemmed from the desire to give their best to their pregnant body as well as the developing fetus. One of the expectants illustrates this point clearly:

‘A pregnant woman thinks a lot about her baby, and she has a great desire to bring him to the best place…

  Now, with a baby on the way, I’m so much more focused on my health and looking after my body. I want to give my baby the best start I can’.(W.H; 29 w pregnant)

It seems that the very fact that expectants were now pregnant encouraged them, especially those who identified themselves as having a low level of health awareness, to place their physical needs at the top of their priorities. As the following expectant put it:

‘…I’m considerate of myself, which I wasn’t before. I am a teacher in a nursery school, and now I treat myself first more than my pupils. For example, I go to the toilet without waiting, even if a pupil has to wait’.(W.A; 15 w pregnant)

**Significance of Nutrition.** Most of the expectants held the opinion that the foods found in their homes were important, especially during their first pregnancy. While the family is in the process of forming and particularly considering cravings for snacks, it is vital to manage the products that are brought into the house. As most mothers said, bringing healthy foods into the home helps to maintain a healthy diet by encouraging better resistance to temptations.

‘What isn’t at home you don’t eat. It’s worth having good things at home to eat, especially now as parents-to-be. If you have loads of pasta, you’ll make pasta and that’s what your child will eat as well. If you have whole rice and quinoa that’s what you’ll all eat…We decided that at home we would have mostly healthy things, to think about what enters our home as a family right from the beginning. We do eat less healthy things outside the house when we go out…’(W.G; 7 m post-pregnancy)

All of the expectants stated that healthy nutrition gives strength in general and particularly during the challenging period of pregnancy. Some added that especially now that they felt exhausted, devoting time to understanding proper nutrition and eating healthy foods is even more important to gaining strength and nourishing both themselves and their baby. As one expectant explained:

‘If you eat correctly in pregnancy, then the pregnancy is easier. It should make it easier if you already come with a healthy lifestyle to breastfeeding and taking care of the baby it all looks different… I feel for myself that when I eat more fruit and vegetables, I feel he is healthy, I feel livelier and more energetic’.(W.E; 23 w pregnant)

**Significance of the Spousal Relationship.** Nearly two-thirds of the expectants emphasized the importance of their partner’s involvement in the matter of nutrition at home during pregnancy. Foremost, they highlighted their partner’s support and concern for a healthy diet. Some brought up the importance of marital preparation for birth and parenthood. They voiced that this preparation is part of a joint and long journey ahead, with nutrition being one of its components. They also noted that it is worth getting used to healthy food as a couple during pregnancy, making it possible to later set a personal example as parents. As the following expectant explained:

‘We, as a couple of parents-to-be, really need to give some thought to the question of what foods we want at home, and be a personal parental example… Now, during pregnancy, it’s really important to him what I eat. He will help me not to eat chocolate. Because I complain a lot that I’m eating junk, so he helps me’.(W.F; 26 w pregnant)

THEME 2: Difficulties in the home food environment during pregnancy.

Most expectants commented that their pregnancy brought with it significant physical and emotional changes. Coping with these challenges affected their dietary choices, especially during the first trimester.

**Physical Symptoms in Early Pregnancy that Challenge Nutrition Choices.** The expectants characterized the beginning of pregnancy as a time of significant physical changes, in the sense of taste, nausea, vomiting, food intolerances, and cravings.

‘The whole time of nausea in the first trimester, I nearly didn’t eat, and then cornflakes, which we hadn’t had before, entered our house. And all sorts of snacks that we would never in our life have bought before too. Chocolate—we always had, but there are situations in pregnancy where we allow ourselves more’.(W.N; 37 w pregnant)

**Emotional Load During First Pregnancy.** Many women, during and after pregnancy, noted that their pregnancy, especially the first and third trimester, was characterized by emotional unavailability that affected their ability to make dietary changes. In the first trimester, the emotional load stemmed from becoming pregnant and recognizing it, along with the medical observations and appointments. Later, preparation for childbirth and anticipated parenthood all lead to an emotional burden.

‘I think changes should be done from the start of the pregnancy but for myself, I think it’s a lot of changes at the beginning of the pregnancy so it’s really hard to set it as a tradition at home… my body is changing and I myself don’t know what will happen next and not even what I want to eat. It can happen that I buy something and a week after I won’t be able to even smell it’.(W.C; 20 w pregnant)

When the expectants were questioned about appropriate interventions for pregnant women, several emphasized that any intervention aimed at benefiting mothers-to-be must consider the challenges of pregnancy. In practical terms, this translates to a straightforward intervention that offers guidance for choosing healthier products. Any more substantial intervention, such as aiming to change eating habits, would not be appropriate. A second suggestion that several expectants raised was that interventions should focus on the couple, rather than on preparing for feeding the unborn child.

‘This understanding, knowledge and skills seem very, very necessary to me. Even though I am someone who has these skills and I grew up in a house that held healthy lifestyle, I would be very happy to be guided about foods for a healthier home. … It has to be simple because you can’t absorb that much information. You also lose so many brain cells during this period, you don’t remember, it has to be very practical, so that it will be feasible, and recipes for example must be easy and accessible…’(W.K; 12 m post-pregnancy)


**THEME 3: Insufficient Guidance Concerning PHFE During Pregnancy.**


During routine pregnancy appointments, the expectants received almost no nutrition guidance, and particularly lacking was guidance concerning the PHFE. The principal nutrition issues that were discussed were folic acid, iron and multivitamins, drinking adequate water, and breastfeeding recommendations:

‘During follow-up checks, we didn’t talk about food at home and things like that. The nurse told me to drink two liters of water a day, and I drink two liters of water, “no questions asked”. But apart from that, nothing’.(W.O; 37 w pregnant)

In the study’s second phase, the mothers expressed regret that they had not prepared their PHFE prior to birth. Some suggested that instruction regarding this would have been a welcome addition to routine check-ups. Planning healthy food availability to set up their PHFE was seen as a game changer. Mothers explained the importance of healthy snacks, freezing cooked meals, and methods for preparing healthy, inexpensive, tasty, and quick foods.

‘There is a lot of need for snacks—to have food available that gives energy, healthy snacks that can be prepared in advance and frozen… I would eat at night so as not to fall asleep while breastfeeding and I had to have a lot of snacks. For example, cutting vegetables when the baby is asleep and then when she wakes during the day, I eat them because otherwise, I don’t have enough time to get to vegetables … ready-to-freeze meals in advance, lots of things to learn and prepare before’.(W.M; 18 m post-pregnancy)


**THEME 4: The Timing of PHFE Guidance Is Crucial.**


The participants were interested in learning about nutrition but emphasized that timing was of pivotal significance. Their physical and mental availability was typically maximal in the middle of pregnancy, when women are at their best physical and emotional state. In motherhood, they pointed towards an additional need, namely continued support. They noted that guidance was most welcome in preparing for the transition to eating solids.

**The Second Trimester of the First Pregnancy Presents an Opportunity to Shape a Healthy Home Food Environment.** Most expectants and mothers indicated that the second trimester is the most suitable time to receive PHFE guidance. In comparison with the other two trimesters, they felt physically better and emotionally and technically available. They emphasized that they would be happy to learn about a healthy PHFE and thought it best if it was included for all first-time expectants in routine health visits.

‘As women, in pregnancy, especially first pregnancy, from the second trimester onward-it’s an excellent opportunity to learn about good foods to have at home… for us now for a healthy pregnancy, and for the baby and us as parents to be. It’s better than after birth because now there is anticipation, planning, and availability, time for it’.(W.I; 31 w pregnant)

**There Is a Need for Guidance Following Pregnancy.** The mothers all indicated that they needed guidance regarding the food environment at home after pregnancy, too. They felt the need to be role models for their children, and yet they felt unsure which foods were best to include, especially around the transition to solids. Some stated that nutrition at home is complex and significant. They needed continuous reinforcement and practical guidance, recognizing that the child would eat what parents were familiar with.

‘Yes, I lack knowledge, how can I combine easily and tastily what he needs, what kind of meal to prepare… I want to set a good example for him. I think that if you want your kid to eat healthy you should eat healthy because he sees and wants what you eat’.(W.L; 11 m post-pregnancy)

It is noteworthy that dietary decisions made during pregnancy may be indicative of a heightened awareness of nutrition during both pregnancy and early motherhood, together with challenges, such as cravings for snacks during pregnancy and time constraints postpartum. Most mothers indicated that the availability of vegetables and fruits within reach decreased after birth.

‘We eat similarly, the three of us… but since he was born, we invest less and less in health and nutrition. Which means—we add a little more salt, less cooked food, more cakes and cookies, much less fruits and vegetables than before… Why? Because then, before we became parents, we had leisure and time. Now less. Although it’s funny with one child you might think. People have 4 children and we have one’.(W.O; 13 m post-pregnancy)

These data are also reflected in Table 2’s PHFE results, indicating a decrease in fruit and a more significant reduction in vegetable availability postpartum. As some mothers shared, the research itself fostered introspection on nutrition at home:

‘You know, the very research made me think about things. When we were shopping, and also cooking—I remembered the questions and all the research—to buy it or not? When cooking, add oil or not. It definitely added to the conscience but also changed the behavior in practice’.(W.F; 7 m post-pregnancy)

## 5. Discussion

The aim of the present research was to examine attitudes and practices of first-time pregnant women pertaining to the physical home food environment and to explore if the first pregnancy provides a teachable opportunity to enhance the PHFE. Our findings suggest that the first pregnancy is a unique period of time with considerable motivation and availability for healthy PHFE changes, while acknowledging the physical and emotional challenges.

The first motif we observed pertains to the attention given to bodily health and nutrition during the first pregnancy. This corroborates existing research that indicates that first-time mothers prioritize health for maternal and fetal well-being compared to subsequent pregnancies, where there is less emphasis [34]. It is worth noting that health orientation prior to parenthood is another factor that appears to impact the direction of changes in women’s diets, as those with previously low food and health orientation report dietary improvements in the transition to parenthood [35].

The second motif observed in this study concerns how physical symptoms in early pregnancy challenge nutritional choices. The high consumption of simple carbohydrates, snacks, and fried foods reflects this tendency. Additionally, an increasing trend in the consumption of low-calorie beverages as well as coffee and alcohol intake in early parenthood was noted. It is well-described that during first-time parenthood, ‘obesity-protective foods awareness’ coexists with an inhibiting factor, the limited time and fatigue experienced during early parenthood, as detected in our research and various other studies [36]. These barriers may lead expectant and new mothers to choose readily available foods, especially under pregnancy-related constraints [37].

Prior research has also emphasized the role of home food availability during pregnancy and postpartum, particularly in low-income households [38]. Concerning the content, the current study suggests the need to focus on methods to enhance the availability of raw vegetables and healthy snacks within the PHFE during the first pregnancy. In contrast to macrolevel applications, it is feasible to engineer microscale physical environments inside of the home to provide individuals with personal ways to control their eating behaviors. By engaging in this guidance, by doing ‘homework’ by themselves before the baby joins the family, parents will be able to serve as better role models in a ‘nutrition-supportive environment’ in shaping their family’s dietary habits over time. Underpinning this is the concept that reducing obesogenic PHFE during the first 1000 days may promote the development of more healthful diets.

A supportive environment is a crucial component of the health promotion paradigm [39]. As nutrition gatekeepers, parents-to-be are capable of handling the availability of foods at home, which will impact their and their children’s eating patterns without excessive restriction. Based on the findings of this study, it is evident that both the content and timing of the PHFE intervention play a significant role, especially during the first pregnancy. Women in their first pregnancy are interested in practical and concise guidance regarding the PHFE, particularly in the middle of pregnancy, and this is boosted at weaning. Integrating these conclusions, along with the nesting effect in the first pregnancy, leads to the inference that molding the PHFE in the first pregnancy provides a broad window of opportunity that opens for a short time. We would like to offer a new concept that encompasses *nesting* in the matter of nutrition: *NESTRITION* or nutritional nesting. This involves guiding expectant parents during their first pregnancy to create a healthy home food environment. It includes practical advice on purchasing and preparing nutritious foods, particularly raw vegetables and healthy snacks, to support better dietary habits as they prepare for the baby’s arrival.

A further concept relates to the identification of discrepancies between needs and availability, which is essential for successful health promotion. Similarly to Tudor Hart’s ‘Inverse Care Law’ [40], which asserts that the provision of good medical care tends to be inversely proportional to the population’s need for it, our research emphasizes a challenge for health promotion and preventive medicine that can be termed ‘The Inverse Carer Law’. Early parenthood is a critical period for prioritizing health promotion, but it is also when parents possess the least capacity to address these needs [41]. The need identified in our research for counseling in the first months of the baby’s life, particularly during the transition to solid foods, has also been observed in other studies within the field [42]. In the context of the current study, this finding underscores the importance of anchoring health promotion messages over time, particularly during the transition to parenthood and while adopting new practices.

Our study and its messages need to be considered in light of some limitations. Firstly, the sample was primarily composed of middle-class women, with little representation of individuals from lower socioeconomic backgrounds. It is likely that the decrease in vegetable and fruit consumption postpartum would be even lower among mothers from more socioeconomically disadvantaged backgrounds. Most participants were also a healthy weight, and the views of women coping with overweight would be of particular value and provide greater depth to the conclusions drawn. In addition, the study was conducted in one country alone. Although this allows for a culturally relevant reference for this specific population, thus enhancing the applicability of the health promotion program in practice, one must be cautious in extrapolating to other populations.

## 6. Conclusions

Our study highlights the potential of nutritional nesting during the first pregnancy as a preventive strategy to establish a healthy PHFE, which is vital for shaping early dietary patterns and reducing the need for challenging dietary modifications later. This approach aligns with primary prevention and nutrition education goals, thus potentially mitigating obesity risk among young parents and their offspring and thereby contributing to the prevention of chronic conditions through early dietary assessment and intervention.

Incorporating targeted guidance on creating a nutritious home food environment into antenatal care is crucial, as current support for first-time mothers remains inadequate. Given the heightened focus on fetal and maternal health during pregnancy, particularly in the second trimester, interventions should be simple, evidence-based, and tailored to the unique needs of expectant mothers. This aligns with the broader goal of dietary assessment in the prevention and management of chronic conditions by promoting healthier dietary behaviors from the outset.

Further research should explore this topic among diverse populations, particularly those from lower socioeconomic backgrounds and at-risk groups, to address disparities in dietary behaviors and chronic disease prevention. Understanding the partner’s role in nutritional decisions during first pregnancies and developing practical, effective interventions to enhance the PHFE are essential steps to ultimately improve health for many in the future.

## Figures and Tables

**Table 2 nutrients-16-03335-t002:** Food frequency during pregnancy and in the year following birth.

How Many Times Do You Eat the Following Foods Each Day? 8-Point Likert Scale (0–6+)		During Pregnancy *n* = 15	Post-Pregnancy *n* = 15
		Mean (SD)	Mean (SD)
Desirable foods	FRUIT	2.2 (1.3)	1.6 (1.2)
	RAW VEGETABLES (e.g., lettuce, tomatoes, salad)	2.3 (1.4)	1.7 (1.0)
	COOKED VEGETABLES, not including potatoes (e.g., carrots, courgettes, broccoli)	1 (0.96)	1.1 (0.63)
	No. of individuals consuming 5+ portions of fruit and veg per day	9 (60%)	6 (40%)
Less-desirable foods	CRISPS OR OTHER SAVORY SNACKS	0.6 (0.8)	0.5 (0.8)
	HIGH-FAT, PROCESSED FOOD (e.g., cream, chips, fried food)	0.9 (1.0)	0.5 (0.8)
	HIGH-SUGAR FOOD (e.g., sweets, cakes, cookies, chocolates)	1.9 (1.0)	1.4 (1.5)
	PROCESSED MEATS (e.g., hot dogs/burger/sausage)	0.5 (1.3)	0.2 (0.4)
	SWEET BEVERAGES	0.9 (1.2)	0.4 (0.8)
	ALCOHOLIC DRINKS	0.1 (0.2)	0.4 (1.3)
	COFFEE/BLACK TEA	1.0 (1.3)	1.5 (1.8)
	LOW-CALORIE/DIET DRINKS	0.2 (0.5)	0.4 (1.0)
PHFE	Fruit accessibility ^1^	14	11
	Vegetable accessibility ^2^	7	4

^1^ Number of women who answered yes to ‘Without opening any opaque cupboard doors, is there any kind of fruit in your home now, displayed out in the open?’ ^2^ Number of women who answered yes to ‘Do you have any ready-to-eat fresh vegetables on a shelf in the fridge or on the kitchen counter now?’

**Table 3 nutrients-16-03335-t003:** Theme descriptions and examples.

Theme		*n* *		Example
		During Pregnancy *n* = 15	Post-Pregnancy *n* = 14	
**THEME 1**	**First pregnancy as an opportunity to improve the home food environment.** **Recognition that pregnancy, diet, and family are factors that can influence the PHFE**.	13	12	*‘In first pregnancy—I think that we, as parents to a firstborn, are thirsty to receive new information…. with the following children it is less so. (Intervention) during pregnancy is better than after birth…’* *(W.I; 18 m postpartum)*
**Sub theme 1.1:**	**Importance of pregnancy.** Women during their first pregnancy are attentive to their body’s needs and tend to lead a healthier lifestyle for their own and the fetus’s benefit.	11	9	*‘I think there’s something in the pregnancy itself …. that the women themselves think about the body and food and there is a lot of preoccupation with these things due to concern for a healthy body during this time’.* *(W.N; 37 w pregnant)*
**Sub theme 1.2:**	**Significance of nutrition.** Importance of deciding what enters the house, especially during a first pregnancy, which shapes the family and when cravings may occur. A healthy diet provides much-needed strength to deal with the challenge of pregnancy.	13	8	*‘It’s true that I don’t have much strength right now but compared with the beginning of the pregnancy I have more. It’s worth spending the little strength I have to know how to get more strength through nutrition, for all of us’.* *(W.I; 31 w pregnant)*
**Sub theme 1.3:**	**Significance of the spousal relationship.** Support from partners during pregnancy in caring about diet and a need for joint involvement in preparing to raise a child.	10	10	*‘I think it’s very important, couples meeting during pregnancy, in preparation for the fact that we are going to give birth to a child… There’s a shared process that happens beforehand, and if it doesn’t happen, you need to make it happen’.* *(W.M; 37 w pregnant)*
**THEME 2**	**Difficulties in the home food environment during pregnancy.** Physical and emotional obstacles of pregnancy that hamper improving the PHFE.	13	8	*‘…in the first three months I suffered from nausea and it was really hard for me to eat. I lost about three kg… I have ready-made Schnitzels in the freezer, which is processed food, and I love it and at the beginning of pregnancy I ate lots of it’.* *(W.B; 14 w pregnant)*
**Sub theme 2.1:**	**Physical symptoms in early pregnancy that challenge nutritional choices.** Women experience changes in their sense of taste, desire for carbohydrates and snacks, revulsion towards food, nausea, and vomiting, especially at the beginning of the pregnancy.	13	8	*‘I have an aversion to meat and fish unless it’s something that I really love, I’m disgusted with making meat and fish. To tell the truth it’s surprising, as I am a ‘meat’ person I can eat meat first thing in the morning. But there are all sorts of symptoms in pregnancy’.* *(W.A; 15 w pregnant)*
**Sub theme 2.2:**	**Emotional load during the first pregnancy.** The first pregnancy is an emotional experience that may create an intolerance for change.	8	7	*‘Up until now, you have been going about things in a certain way, and then, during pregnancy, which is an unconventional time that increases anxiety, requires a lot of comforts, and takes you out of your comfort zone—are you going to intentionally make a change? It’s very hard to create a change on top of change, it needs utmost precision’.* *(W.K; 12 m postpartum)*
**THEME 3**	**Insufficient guidance concerning the home food environment during pregnancy.** Women in their first pregnancy do not receive adequate guidance about the PHFE—advice mainly focuses on dietary supplements, drinking fluids, and breastfeeding.	14	11	*‘Absolutely no one spoke with me about nutrition and food at home, nothing at all. Even things that I already know for instance cabbage during breastfeeding with colic and so on… I don’t need it, but it’s sad, it’s a shame…’* *(W.K; 33 w pregnant)*
**THEME 4**	**The timing of guidance is crucial.** Guidance during the first pregnancy is important, but it is better if it is provided at the peak of mental and physical availability in the middle of pregnancy and when the child starts feeding themself.	14	13	*‘Recommendations can only be received when it’s relevant to me and when I’m available for it. It should be the simplest and easiest thing to implement… at the right time it will give me tools, and I will remember and not forget’.* *(W.J; 7 m postpartum)*
**Sub theme 4.1:**	**The second trimester of first pregnancy presents an opportunity to shape a healthy home food environment.** Considering the physical and emotional improvements during the 2nd trimester, a unique window of opportunity opens for shaping the PHFE in the first pregnancy.	13	12	*‘Look, what we as parents bring into the house and eat, that’s what our baby is eating… That’s why one should get guidance like this during pregnancy, in the middle of it, after all the unsteadiness of the beginning and you feel pregnant already and better. Then I would know everything at the start-point and get used to it before the birth and the marathon of parenthood’.* *(W.G; 7 m postpartum)*
**Sub theme 4.2:**	**There is a need for counseling following pregnancy, especially around the transition to solids.** PHFE guidance requires continued input postpartum, especially around the time of introduction to solid foods.	8	14	*‘Around the start of feeding the first child, the transition to solids, there’s motivation and need again, especially with the first child—for example, I see how other mothers invest the energy to make little pancakes from cottage cheese’.* *(W.M; 18 m postpartum)*

* Number of informants who alluded to the theme.

## Data Availability

The original contributions presented in the study are included in the article, further inquiries can be directed to the corresponding author.

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
