# Peer review of "Nutritional Nesting (Nestrition): Shaping the Home Food Environment in the First Pregnancy"

_nutrients, 2024, doi:10.3390/nu16193335_

Round 1
Reviewer 1 Report
Comments and Suggestions for Authors
Thank you for the opportunity to review this manuscript, which describes a longitudinal mixed-methods study designed to better understand primiparous birthing individual’s knowledge, attitudes, and practices related to the physical home food environment (PHFE) and to begin to understand how interventions might enhance the PHFE during this critical period. The manuscript covers an important topic that is relevant to the readership of this journal. Other strengths include the novelty and importance of this study; longitudinal design; rigorous qualitative methods; and generally clear and concise writing. Despite these strengths, some limitations should be addressed. These are outlined in detail below, but major themes include: ensuring that language is inclusive; providing more detail on some of the Methods; being careful not to imply statistical significance in the quantitative results summaries given that no inferential tests were conducted; more information/detail about what was gained from the postpartum interviews. Overall, I believe this manuscript makes an important contribution and will be strengthened with some additional edits.
Language considerations throughout:
*You use the terms women and mothers throughout, which may be appropriate but I cannot tell if you actually assessed gender identity in your study and/or if the studies you refer to in your introduction/discussion assessed gender identity. You may want to change your language (e.g., to “pregnant person” or “birthing person”) or at least acknowledge that people who are pregnant/give birth may identify differently than women or mother. For a longer discussion of gender-related language in research on pregnancy see:
Rioux, C., Weedon, S., London-Nadeau, K., Paré, A., Juster, R. P., Roos, L. E., ... & Tomfohr-Madsen, L. M. (2022). Gender-inclusive writing for epidemiological research on pregnancy. J Epidemiol Community Health, 76(9), 823-827.
*I suggest using person-first language throughout, which you mostly do but there may be a few places that could be modified. For example, on line 72 instead of “obese or disadvantaged women” say “women with obesity" and perhaps define what you mean by disadvantaged – living below the poverty line? Marginalized in some other way (e.g., based on racism)? Something else? I like the APA’s guide for bias-free language for suggestions about wording some of these things in specific, accurate, and non-stigmatizing ways: https://apastyle.apa.org/style-grammar-guidelines/bias-free-language
Introduction: The introduction is strong, includes relevant literature, and is well-written. I do not have any major suggested changes. Minor changes include:
*I did not understand what was meant by this sentence: “This paper is the first in a series of articles involving both xxxx and xxxx subjects, and the baby’s father.”
Method:
Note: I don’t see Supplementary Material 2 in the submission, so I was not able to read over the questionnaires included and apologize if any of my questions/comments below would be answered by that document!
*I may have missed this, but you mention the interviews being translated into English – what language were they conducted in?
*Please specify if there were timeframes in pregnancy and postpartum when participants were assessed, which is particularly important given the amount of biopsychosocial change that happens during these periods.
*Which sociodemographic characteristics were assessed and how were they assessed? I see now that this is included in Table 1, but it would also be helpful to include in the Method - they may also be in Supplementary Material 2, but I think they need at least a mention in the Method text. I am also wondering how BMI was calculated - were participants weighed/height measured in person or was this self-report? Was pre-pregnancy BMI available? Or gestational weight gain?
*Did you consider assessing household food insecurity and participation in any food resource program for low-income families? These factors seem particularly pertinent to the PHFE.
*Could you clarify what you mean by “the transcription translated” (line 119) - I assume this means translated to English but I’m unclear on what the initial language was, and whether the lead research has fluency in both languages or experience with translation.
*What software, if any, was used to assist with the qualitative analysis?
*Were interviews conducted in pregnancy and postpartum pooled to identify themes, or were these considered two batches of interviews, with qualitative analysis proceeding separately for each batch? Please provide justification either way.
*Please provide citations that support the foods that were categorized as healthy and unhealthy (in the Quantitative Analysis section).
Results:
*How are you defining “healthy” (line 193)? Is this related to the healthy lifestyle item? If so, I would reword to indicate that this is the participants subjective report.
*Looking at Table 1 – was ethnicity assessed/anything about cultural background assessed? Was gender assessed?
*Although I agree that it was appropriate not to conduct formal statistical analyses of quantitative data with such a small sample size, I think that you therefore cannot comment on changes in quantitative measures from pregnancy to postpartum (lines 206-215). OR you can present the mean different between pregnancy and postpartum in Table 2, but be clear in text that you are not assessing whether differences are statistically significant.
*It seems like most of the qualitative results discuss “expectants” which leads me to believe that most of the data were coming from the pregnancy interview? I am just curious if you could provide more context for what the postpartum interviews did or did not add. Were participants’ perspectives pretty much the same in pregnancy as in postpartum or was there new information gained from the second interviews?
Discussion/Conclusion
*These sections are written clearly and I agreed with the limitations listed; I also love the term Nestrition!
Reviewer 2 Report
Comments and Suggestions for Authors
Dear Authors
The research is interesting, most of the description is correct, but some details require clarification and correction:
Methods
Line 75: Are these xxxx designations intentional?
1) In which country was the research conducted?
2) Line 90, line 120 : Are these (xx) designations intentional?
Results
1) Under the table 1 Authors explain some scales:
1 ”What is the highest level of education you completed?” … 5=High school. 6= College/University. 199
2 “What is your monthly household income before tax and including benefits (converted to $). 200
3 “Overall, how healthy do you think your lifestyle is at the moment? 1= not at all to 10= very healthy 201 (Howlett et al., 2021). 202
4 “How satisfied are you with your health?” 1.Very dissatisfied. to 5. Very satisfied (Skevington 203 et al., 2004).
but in my opinion this should be explained in study methods section.
In this sentence: Consumption of fruit and vegetables in pregnancy was relatively healthy with most 206 participants eating >5 portions/day (see Table 2).
I recommend changing the phrase "relatively healthy" to "mostly in line with recommendations."
2) Table 2 - Is the superscript "1" marked correctly?
Reviewer 3 Report
Comments and Suggestions for Authors
The article ‘Nutritional Nesting (’Nestrition‘)- Shaping the Home Food Environment in First Pregnancy’ presents interesting issues related to the impact of the first pregnancy on shaping the home food environment (PHFE). The study investigated first-born women's knowledge, attitudes and practices regarding PHFE and assessed whether the first pregnancy presents an opportunity to improve PHFE in newly formed families. The paper is largely based on qualitative analysis using interviews and questionnaires.
The main strengths of the article are the solid theoretical grounding, the rich qualitative analysis and the attempt to define the new concept of ‘nestrition’ - a combination of the concepts of nesting and nutrition. The article also recognises the importance of partner support and the physical and emotional challenges of pregnancy.
Nevertheless, the article has several important limitations that should be taken into account before publication. Firstly, the research sample is small and homogeneous, consisting mainly of women of middle socio-economic status and healthy weight. There is a lack of representation of women from lower social strata and those with overweight problems, which greatly limits the generalisability of the results to the wider population. Secondly, the study was conducted in one country, which limits the applicability of the findings to other cultural contexts. Also, quantitative analyses are limited and there is a lack of meaningful statistical comparisons between different study periods, which weakens the strength of the conclusions regarding changes in eating behaviour after childbirth.
I recommend that significant improvements be made prior to publication. In particular, it is necessary to extend the study to a more diverse sample, taking into account women from different socio-economic backgrounds and with different body mass indexes. More sophisticated statistical analyses should also be carried out to make full use of the data collected. Without these amendments, publication of the article in its current form may not meet the standards required for a scientific journal, especially in the context of the overall conclusions the authors are trying to draw. However, if these corrections are taken into account, the article could be a valuable contribution to the field of maternal nutrition and health research.
